# Structural Monitoring of a Large-Span Arch Bridge Using Customized Sensors

**DOI:** 10.3390/s23135971

**Published:** 2023-06-27

**Authors:** Isabelle Ietka, Carlos Moutinho, Sérgio Pereira, Álvaro Cunha

**Affiliations:** CONSTRUCT—ViBest—Faculty of Engineering (FEUP), University of Porto, R. Dr. Roberto Frias S/N, 4200-465 Porto, Portugal; moutinho@fe.up.pt (C.M.); sbp@fe.up.pt (S.P.); acunha@fe.up.pt (Á.C.)

**Keywords:** structural health monitoring (SHM), operational modal analysis (OMA), traffic loads and temperature effects on structures, arch bridges, customized sensors

## Abstract

Due to the increasing importance of the continuous monitoring of Civil Structures, this research aims to take advantage of new solutions of measurement systems and sensors in the Structural Health Monitoring of bridges, using the reinforced concrete arch Arrábida Bridge as a case study. With the support of customized sensors, this work starts by performing preliminary ambient vibration tests on Arrábida Bridge, aiming at the identification of the natural frequencies and respective vibration modes of the deck. Then, the measurement campaigns carried over time are described, which involved different types of customized sensors, namely, accelerometers, temperature sensors and displacement sensors. Based on the signals collected by these devices, some preliminary analyses were performed. The results show that the temperature measured at the deck sections presents different amplitudes and phase shifts when compared to the temperature measured at the arch sections. Moreover, using the temperature measurements, it is possible to estimate with good accuracy the displacements in the expansion joints of the bridge. It was also observed that the displacements in these joints, although being conditioned by the temperature effects, are also marked by a dynamic component arising from the traffic loads over the deck. The observation of this phenomenon is an innovative aspect found in this investigation, which can be used in the future to characterize the traffic loads on the structure.

## 1. Introduction

The experimental evaluation of the behaviour of structures has been a sustained pillar for the development of structural engineering. Observation-based experimental methods are associated with the first approaches to structural analysis and design. This could result in empirical formulations that served as a basis for designing similar structures [1].

Even with the emergence of major analytical formulations, the credibility of observation in structural analysis and safety assessment is a practice that has and will continue to have great importance in the development of Structural Engineering [2]. In this context, the role of several private and public institutions throughout the world in terms of structural testing and monitoring is emphasized.

The evolution of measurement systems in recent decades is associated with the development of computational analysis tools that have made it common practice to implement continuous monitoring systems in relevant Civil structures, with the following benefits [3]:Verification of calculation assumptions with potential benefit in improving analysis criteria and designing similar structures in the future;The timely detection of possible structural deficiencies, damages or accidents, allowing for increased levels of safety in general;Sustained planning of interventions, such as rehabilitation and structural reinforcement works, according to the effective needs of the structure;Assessment of the effectiveness of maintenance and reinforcement interventions carried out over the life of the structure;Evaluation of the structural conditions in real time immediately after accidents or extraordinary requests;The quantification of the real actions involved in the structure, such as traffic characterization and the quantification of wind actions;Evaluation of the performance of new materials and/or new structural systems.

In the past, the evaluation of the structural conditions of Civil Structures essentially involved the monitoring of static or quasi-static variables. An example of this is the assessment of excessive deformations in structural elements, the measurement of cracking phenomena in reinforced concrete sections, corrosion in rebars, and fatigue tests in existing critical elements [4].

More recently, dynamic testing techniques have assumed great importance in Structural Engineering, driven by the development of Experimental Modal Analysis (EMA) tools, which are based on the estimation of a set of Frequency Response Functions (FRFs) relating the applied force and the corresponding response at several pairs of points along the structure, with enough high spatial and frequency resolution [5,6,7].

Due to the fact that applying dynamic loads to structures can be a difficult task, Operational Modal Analysis (OMA) tools started to gain more popularity, particularly in relation to large Civil Structures [8,9]. In fact, they allow for the continuous identifyication of the modal parameters of a structure under operating conditions, taking into account the measured ambient vibrations without needing to stop the traffic. The main advantage of OMA is the possibility of detecting small changes in some modal properties of the structure, being an indicator of the occurrence of structural damage [10,11]. This way, systems are observed as a whole, making it possible to detect the existence of damage without resorting to generalized instrumentation, resulting in economic benefits.

Whether it is static or dynamic monitoring, there is a current trend in developing new measurement devices and sensors, allowing for evolution toward a new generation of instrumentation systems [12,13]. In particular, the difficulties that are to be overcome in traditional data acquisition systems (DAQs) include [14]:The high cost of traditional DAQs;Excessive dependence on an external power supply;Frequent failures in data acquisition attributed to heavy operating systems;Difficulty in being physically integrated into structures, such as taking up too much space and being barely manageable;When using cables, the number of sensors is limited due to the exponential complexity of the acquisition system.

In contrast, data acquisition systems based on new sensors offer the following advantages [15]:Cost-attractive;Enable wireless solutions;Autonomous and small size (easily integrated into structures);Possibility of installation of a high number of sensors;Low power consumption (can run on batteries or solar panels);Simplicity of functioning and robustness;Low maintenance;May be easily customized.

The use of sensors with these characteristics makes it possible to trivialize the implementation of sensors in structures in such a way that the difficulties associated with expensive and complex systems are overcome. However, it is worth mentioning that traditional systems still have their role in structural monitoring, particularly when dealing with very stiff structures with very low energy signals and when superior performance in terms of accuracy and data throughput is demanded.

In this context, the main objective of this paper is to investigate and show how a more accessible sensor technology may be used to achieve the same results as a more complex monitoring system. As such, the developed sensors are detailed in terms of their functioning and components, and their implementation to a large arch bridge is described in a monitoring plan. The effectiveness and usefulness of these sensors are demonstrated through the presentation of some preliminary analyses that can be performed recurring to the signals collected over approximately one year of measurements. Because it was possible to measure the displacement in the expansion joints, the observation of a strong dynamic component was observed, which will allow the development of a future investigation into how this phenomenon can be used to characterize the traffic loads on the structure.

## 2. Developed Customized Sensors

The devices installed at Arrábida Bridge for structural monitoring are composed of autonomous modules that do not require wiring scattered throughout the structure. This advantage greatly facilitates the logistics and the process of installing the sensors. In addition, they have great autonomy despite being powered by batteries.

Another important advantage of these sensors is their reduced dimensions, allowing them to be integrated into the structure with no visual or architectonical impacts. This also makes it possible for them to be safe from acts of vandalism.

### 2.1. Accelerometers

Accelerometers are one of the most common devices in structural monitoring [16,17,18]. The modules used in this work are composed of an Arduino-based microprocessor that controls the entire process of data acquisition. The adopted accelerometer is of the MEM type (micro-electro-mechanical), namely the ADXL355 from Analog Devices Company. This digital tri-axis accelerometer presents some interesting features, such as the relatively low noise and the 20-bit Analog-to-Digital resolution. It has a programmable sampling rate and programmable band-pass digital filters. In this application, the sampling rate was set to 62.5 Hz, and the low and high cutoff frequencies of the band-pass filter were fixed at 0.0024 Hz and 15.625 Hz, respectively, allowing for the characterization of a significant number of the first natural frequencies of the bridge.

The data collected by the accelerometer’s modules are saved locally on a microSD card. The files have a duration of 10 min each and are organized in a folder structure identified according to the time and date of the measurements.

No wireless data transmission is available, saving battery power and guaranteeing the high autonomy of the device. The model used for the Arrábida bridge accommodates five lithium batteries of the 18,650 type (a little larger than the AA type). The nominal voltage of each battery is 3.8 V, and the capacity is 3400 mAmps, allowing for an overall autonomy of 3.3 months. The synchronization of the acquired signals between the different accelerometer modules is guaranteed through the use of GPS. This module was previously tested against a commercial one in order to verify its correct functioning [19].

Figure 1 shows some photos of the accelerometer module used for Arrábida bridge. The external enclosure has an IP65 and is made of polycarbonate material. The box is fixed to the structure by means of its flanges to a pre-installed flat base. To change batteries and SD cards, the module is removed from its location, and after replacing these components, it is installed back in place. This operation takes about 10 min for each accelerometer.

### 2.2. Temperature Sensors

The temperature sensors are the DS18B20 model from Maxim Integrated Company. These digital sensors measure temperatures from −55 °C to +125 °C, with a resolution of 0.125 °C and an accuracy of ±0.5 °C from −10 °C to +85 °C, which is adequate for general applications in structural monitoring.

The major advantage of this sensor is related to its simplicity because no extra hardware is required and because it has a three-wire communication with the microcontroller. In addition, due to the unique 64-Bit address, a high number of sensors can be used simultaneously through a single digital port on the microcontroller.

In this application, these sensors were inserted 15 cm inside the concrete sections and then wired to the data acquisition system installed inside a polycarbonate box (of the same size as the accelerometer modules). The temperature readings are taken with a periodicity of 10 min, and the data are saved locally on a microSD card. No wireless data transmission is available.

Because the system is in sleep mode practically all of the time, only three batteries of the 18,650 type are required to guarantee an autonomy of at least 1 year of measurements.

Figure 2 shows a photo of the temperature sensor inside a stainless-steel capsule, as well as a photo of one instrumented section.

### 2.3. Displacement Sensors at Expansion Joints

The displacement sensors used to measure the static and dynamic movements of the expansion joints are of the capacitive type and work inside a stainless-steel cylinder enclosure fixed at one side of the joint. It has a shaft that must be connected to the other side of the joint in order to measure the relative displacement between the two points.

The electronics of the data acquisition system are all installed inside the cylinder, as well as four batteries of the 18,650 type used to power the system, which may work in two different selectable modes, namely, static mode and dynamic mode. The static mode is directed to measure static or slow movements of the joints, taking one measurement every 10 min. This way, it is possible to keep the system working for long periods without battery replacement (typically more than 1 year). On the other hand, the dynamic mode uses a sampling frequency of 5 Hz, which allows for the measurement of the dynamic components of the joint movements. However, in this case, the autonomy is limited to 3 months of operation with the use of four batteries. So far, only the dynamic mode has been used in the case of the Arrábida bridge.

One of the significant features of this sensor is related to its ability to take measurements with 0.01 mm precision in a range from 0 to 2 m. In practice, to prevent the sensor from being too long, the measuring range is adjusted according to the expected displacements, often measuring in the range of 0–150 mm, which was adopted in this case.

Figure 3 shows two photos of the displacement sensor installed in this structure. As can be seen, after installing the sensor, mechanical and visual protection is used to prevent or minimize acts of vandalism, as the sensors are accessible from the deck.

## 3. Arrábida Arch Bridge

### 3.1. Historical Aspects

This imposing work of art [20], Arrábida Bridge (see Figure 4), was the second railway bridge connecting Porto and Vila Nova de Gaia, it was inaugurated on 22 June 1963, and its construction was justified by the growing road traffic that had already saturated the existing Luiz I Bridge [21]. Consisting of a span with a total length of 270 m, on the inauguration date, this structure was classified as the reinforced concrete arch bridge with the largest span in the world [22].

In March 1952, the preparation of the preliminary projects for this structure was awarded to the famous bridge engineer, Professor Edgar Cardoso, and his project was approved in 1955.

In 2003, the bridge underwent inspection and repair works that included structural rehabilitation of the reinforced concrete elements that were degraded, repairs to the deck, the replacement of expansion joints, the renewal of the electrical and communication installations and general repainting [23]. As a formalization of the grandeur and historical value that this important structure imposes, the Arrábida Bridge was classified as a Portuguese National Monument on 23 May 2013.

### 3.2. Description of the Bridge

This structure is made of reinforced concrete and presents a total length of 493.2 m and an arch with a horizontal length of 270 m and a 52 m height.

A total of 12 longitudinal beams compose the deck of the bridge, which has a transversal length of 25 m (Figure 5) and longitudinal spans of 21.2 m (Figure 6). The transversal alignments of the deck are supported by four columns through transversal beams, and these columns are supported by two twin arches with an 8 m length each and a bi-cellular box girder cross-section (Figure 7 and Figure 8).

The twin arches present four different types of cross-sections, which are located in the respective alignments: the mid-span of the bridge (center of the arch, section K–K′), the remaining spans (section L–L′), the column support areas (section M–M′) and the end joints connected to the main columns through the foundation massifs (section N–N′). Each section can be visualized as shown in Figure 8.

In addition to the columns supported by the twin arches, the bridge is supported by two pairs of main pilasters, which are located on each side of the bridge and are connected at the base to each extremity of the arch, as shown in Figure 9. These pilasters are transversally connected by special beams that are significantly more robust than the remaining ones and present as a cross-section, as indicated in Figure 10.

On the construction of the bridge, two different classes of concrete were adopted. A concrete composed of 500 kg of cement per cubic meter of concrete, presenting a minimal compression resistance of 400 kg/cm^2^, which corresponds to the actual class, C40/50, was used in the construction of all of the reinforced elements and areas close to the foundation footing and massifs of the arch supports. For the construction of all the footing and massifs elements, the concrete adopted is equivalent to the actual class, C30/37, presenting a composition of 300 kg of cement per cubic meter of concrete and a resistance of 300 kg/cm^2^.

The steel used in the reinforcement corresponds to class S235, with nominal cover varying from 3 cm in the deck elements and 4 cm in the arches and columns.

### 3.3. Ambient Vibration Tests

In the ambient vibration tests, 14 new-generation accelerometers, duly synchronized, were used, which allowed for the completion of the test in two setups, each one with a duration of 30 min.

The measurement sections are indicated in Figure 11, where four sensors were positioned at fixed reference stations (sections 2M, 14J, 4M, 16J; M, upstream; J, downstream) and the remaining 20 units at mobile stations.

The registered signals were processed using the “Peak-picking” method, which permitted the identification of the first natural frequencies and respective vibration modes of the deck, as indicated in Figure 12 and summarized in Table 1. In these graphics, the blue points correspond to the modal coordinates identified on the upstream side of the deck and the orange points on the downstream side. The full line represents the approximate shape of the expected modal configuration.

As can be observed, the first natural frequency is 0.72 Hz, corresponding to a lateral vibration mode. The vertical modes follow the typical flexion modes of symmetrical structures, and the torsional modes present natural frequencies of 2.89 Hz and 3.56 Hz.

## 4. Instrumentation Plan

### 4.1. General Aspects

The installation of the sensors on the bridge was divided into two different phases: the first one in May 2021 and the second in June 2022.

In the first phase, four accelerometers, four temperature sensors and two displacement sensors were installed. In the second phase, one displacement sensor had its position changed, and an additional four temperature sensors were installed. The location of each device will be explained in this section.

### 4.2. Location and Implementation of the Sensors

Regarding the accelerometers, all of the sensors were installed in the first phase of instrumentation and placed on the north side of the bridge, and these devices were named as follows: 2M, 4M, 6M and 16J. The devices with a code ending with “M” were placed upstream, while the ones ending with “J” were placed downstream. This configuration was chosen in order to capture the dynamics of the first vibration modes of the structure (see Figure 13). Sensors 4M and 16J were positioned at the same point in the elevation view but on opposite sides of the deck, enabling better identification of the torsional vibration modes of the bridge.

The installation of the temperature sensors was divided into two phases, with four of them being initially installed in the transversal structural beams below the deck of the bridge, located at the north joint (T1M, T2J) and near the mid-span of the bridge (T3M, T4J). The remaining four sensors were installed in the second phase close to the base of some of the columns supported by the arch: three of them on the upstream side (T5M, T7M and T8M) and one of them on the downstream side (T6J), as shown on Figure 14.

Finally, the displacement sensors were both initially installed on the north joint of the bridge, one on the upstream side and the other on the downstream side, followed by an initial measurement of 10 min. The observation of these signals allowed for the immediate identification of significant dynamics in joint movement, a reason which led to the installation of the second sensor D2(J) on the downstream side in the same north joint. This decision was made in an attempt to find out if the bridge would present lateral movements that could be registered through these devices, which may arise, for example, from the dynamics of the first lateral vibration mode exposed to wind excitations. However, after some months of measurement, it was concluded that no significant relative displacement was detected between the devices installed on each side of the north joint. In other words, the bridge did not present considerable lateral movements that could be registered by duplicate devices on the same joint. For this reason, the second phase of instrumentation consisted of changing the location of the sensor placed downstream of the north joint to the south joint of the structure on the upstream side, now with the code of D2b(M), as shown in Figure 15.

## 5. Preliminary Analysis of the Results

### 5.1. Temperature Analysis

In Figure 16, the annual temperature variation recorded by the sensors installed on the deck (T1 to T4) is visualized.

The analysis of this graph permits the identification of higher temperatures in summer, followed by decreased temperatures in winter. Some extreme temperature events may also be observed during July 2022, despite the limited period shown.

Some statistical values are presented in Table 2, such as the maximum, minimum and average measured temperatures during the period shown.

The outputs obtained through the analysis of the temperature results can be separated into three different components: daily variations, trend variations and annual variations.

As shown in Figure 16, the daily variation and annual variation are easy to visualize. However, the temperature profiles are also marked by a trend component that depends on some daily fluctuation of more or less warm or cold days. It was concluded that, to numerically model the temperature variations, it is necessary to separate the trend component and the daily component, in that the annual variation was coupled to the trend component. This may be performed by applying adequate filters to the temperature signals.

The trend component of the temperature may be obtained by filtering frequencies equal to and higher than the daily component and allowing lower frequencies to pass, which corresponds to the use of a lowpass filter (see Figure 17), while the daily temperature variation may be obtained by using a highpass filter with a cut-off frequency set to a value slightly below the daily frequency (see Figure 18).

On the other hand, Figure 19 and Figure 20 depict the measured temperatures obtained from the sensors installed on the deck (T1 to T4) and the arch (T5 to T8), respectively, during the month of August 2022. These graphs permit the verification that the devices installed on the deck present lower and smoother daily amplitudes. This is because these devices are installed on the transversal beams below the deck, where the sunlight incidence is not made directly on these sections. By contrast, temperature curves of the sensors placed at the arch show higher daily amplitudes and sharper peaks, as these devices are installed on sections of the arch exposed directly to sunlight incidence.

In addition, Figure 20 permits the verification of the peculiar behaviour of sensor T7, which presented lower daily amplitudes in terms of temperature when compared to the remaining sensors installed on the arch. This difference can be justified by considering the section where this device is installed, which is very near to the mid-span of the bridge, where the vertical distance between the arch and the deck is very small, and consequently, this sensor is under the shadow of the deck, receiving less sunlight incidence than the others.

Moreover, the curves of sensors T5, T6 and T8 regarding the period of August/2022 reveals another important point to be discussed, which is also associated with the higher exposure to the sunlight incidence of these devices. In fact, very often, their daily variation curves present two consecutive peaks in short periods of time. This can be explained as a consequence of the changing position of the sun in relation to the position of these sections of the bridge. The first peak happens when the sunlight falls on the measured section with an angle that causes the initial temperature peak. At that time, the section is not under the shadow of the deck. Then, when the sun is over the deck (close to noon), the section becomes covered by the shadow coming from the deck, and so the temperature decreases. After some time of shadow, the section is again directly exposed to sunlight incidence, presenting with a new peak, which is higher than the first one. It is worth remembering that the temperature sensors are inserted into the concrete at a depth of 15 cm. Therefore, the measured temperature is not superficial.

After dividing the recorded temperatures into two groups (deck sensors and arch sensors), the average temperature of each group was calculated, and each curve is represented in Figure 21.

This graph permits the visualization of the fact that the two groups present different phase shifts between them, which was evaluated at −5.33 h, meaning that the temperature sensors in the arch group achieve their peak of amplitude 5.33 h before the sensors in the deck group. This happens due to the placement of the sensors in these two different elements of the bridge. The sensors installed on the deck are located on the transversal beams below the top slab, and so they are not directly exposed to sunlight incidence. Once the light falls initially on the superior part of the deck slab, the heat is conducted through the concrete element from the deck to the beam, taking more time for the sections below the deck to reach the peak when compared to the incidence of sunlight on the sensors placed at the arch, which are directly exposed to it, and for this reason, achieve peak temperature sooner.

### 5.2. Displacement Sensors at Expansion Joints

In Figure 22, the measurements of the displacement of the north expansion joint are shown, which were obtained by averaging the signals from the sensors installed on the upstream and downstream sides. During the periods from 6 November 2021 to 9 November 2021 and 27 December 2021 to 11 January 2022, there was a lack of data from both sensors due to a delay in battery replacement, leading to battery failure in the system during those periods.

The statistic values of the shown displacement variation are summarized in Table 3. It is worth mentioning that zero displacement is stipulated as an initial reference value from which relative displacements are measured.

When the displacement is positive, it means that the sensor is moving forward, approaching the abutment; thus, the bridge presents expansion behaviour. In opposition, when the sensor moves backward, away from the abutment, this indicates that the bridge is retracting in length, corresponding to a negative displacement.

The annual analysis permits the verification of the trend behaviour of the expansion of the bridge for positive displacements during the summer season, characterized by higher temperatures. During the winter season, the trend is in a negative direction, corresponding to the retraction of the deck.

As in the temperature sensors, the displacement of the expansion joints can also be divided into different components. However, in this case, beyond the daily, trend and annual components, a dynamic component can be observed that is only observed on an amplified scale.

This was possible due to the adoption of a higher sampling frequency in the displacement sensors. While the temperature sensors were programmed to acquire one sample every 10 min, the displacement sensors use a sampling frequency of 5 Hz. Figure 23 shows an example of a displacement signal where dynamics with very small amplitude are clearly visualized (in the order of 1 mm peak-to-peak).

Further analysis of this component permits the identification of some relevant aspects regarding the behaviour of the expansion joints. One of these aspects is the significant difference between the displacement records taken in periods of low and high traffic of vehicles over the bridge, as shown in Figure 24 and Figure 25. Here, the signals were acquired at 00 h 00 min and 08 h 00 min, respectively, on a business day on the north joint (upstream and downstream).

It can be observed that the energy contained in the signals is much higher in the period of 08 h 00 to 08 h 10 min when compared to the period of 00 h 00 min to 00 h 10 min. This is attributed to the intensity of traffic flow, meaning that horizontal displacements in the expansion joints are related to the vertical loads applied by vehicles.

This conclusion is reinforced by analysing the root medium square (RMS) values of the joint displacements evaluated in 10 min periods during 1 month of records, as shown in Figure 26. Higher values correspond to daytime periods of more intense traffic, and the lower values correspond to nighttime periods; therefore, it is possible to identify working days, weekends and holidays, such as 3 and 10 June.

Curiously, when sensor D2 was moved to the south joint, the measurement records yielded the outputs indicated in Figure 27. They have similar amplitudes but opposite directions, meaning that when one joint expands, the other retracts, i.e., the deck is moving longitudinally as a rigid body.

The effect of moving traffic on the displacement of the joints may be seen as suggested in Figure 28. Due to the deformation of the arch, when a certain vertical loading is more concentrated on the first half-span of the arch, the deck moves in the right direction, and after the loading changes to the second half-span, the deck moves in the left direction. That is why the signals of the displacement joints depicted in Figure 24 and Figure 25 are marked by sinusoidal or pseudo-sinusoidal motions, which reflect the continuous motion of the loading from one pilaster to another.

In addition, the frequency identification of these sinusoidal motions may be used to estimate the velocity of the vehicles over the bridge. As an example, Figure 29 shows a zoomed image of one of these sinusoidal signals registered during rush hours, where a period of T = 11 s was estimated. Taking into account that the span of the arch from one pilaster to another is 270 m, the velocity of the vehicle is estimated to be 90 km/h, which is reasonably consistent with the normal velocity of traffic over the bridge during that period (see Figure 30).

The previous study focused on temperature measurements, particularly on the phase shifts between several sensors, which was extended to displacement measurement in the expansion joints, with the aim of defining the correlations between both variables. Considering the period from September 2021 to May 2022, the average temperature of the group of sensors T1 to T4 was calculated, and its phase was obtained regarding the average displacement of the sensors installed upstream and downstream (D1 and D2). This analysis was performed for both daily and trend components of the results obtained by both temperature and displacement sensors.

The phase shift of the daily component of the temperature signal regarding the displacement was found to be −2.5 h, and for the trend component, the phase was zero, meaning that both temperature and displacement trends are in-phase.

As explained before, the negative signal indicates that the temperature sensor is ahead of the displacement sensor, meaning that the temperature achieves its peak amplitude before the displacement. The positive signal denotes the opposite situation.

To find the correlation between the daily component of temperature and displacement, first, the temperature signal was aligned in order to be in the same phase as the displacement signal. Then, the temperature-versus-displacement chart was plotted, and its linear correlation line was obtained, as shown in Figure 31.

Regarding correlation between the trend component of temperature and displacement, the signal alignment is not necessary as they have the same phase shift. Therefore, in Figure 32 the temperature versus displacement chart is plotted and the linear correlation was found.

Even though both correlation factors R are not very close to unity (although the correlation factor in the case of the trend components is higher than in the case of the daily components), it should be possible to take advantage of these correlations in order to estimate joint displacements from temperature measurements (and vice versa).

In fact, knowing that the total values of both variables may be obtained by summing the respective trend and daily components, the estimate of the total displacement in the expansion joints, according to the obtained numerical correlations, may be estimated as follows:(1)$$D=6.3328\cdot T_{DC}+0.0007+2.4502\cdot T_{TC}-29.564\;D=6.3328\cdot T_{DC}+2.4502\times T_{TC}-29.563$$
where:

D is the estimated total displacement;

$T_{DC}$ is the daily component of the measured temperature;

$T_{TC}$ is the trend component of the measured temperature.

As a result, Figure 33 shows a comparison between the measured displacement (D_measured) at the joints and the estimated displacement (D_estimated) using the temperature measurements. Despite the previous comments related to correlation factors, the approximation between the two curves is very good. This means that by measuring the temperature in the appropriate sections of the bridge, it will be possible to estimate with good precision the displacements in critical sections of the structure.

### 5.3. Identification of Modal Properties over Time

The identification of natural frequencies and damping factors in the time domain proceeded through the use of operational modal analysis (OMA) tools, which involves the estimation of modal information from structural responses (outputs) recorded while the structure is under normal operation conditions [24]. Since no specific excitation is purposely induced on the structure, and the modal identification relies solely on structural responses and on the assumption that the input is white noise, the methods used in OMA are commonly known as output-only methods [25,26].

This process was realized using the measurements recorded from four accelerometers installed on the bridge deck, as explained before. Each device permits the recording of accelerations in longitudinal, vertical and lateral directions. However, in this study, more relevance was put on the vertical and lateral results, as the preliminary ambient vibration tests revealed that the structure does not present significant longitudinal movement dynamics (except for the 1st vertical bending mode, where a small longitudinal component is also present).

A first check on the signals was performed by calculating the root mean square (RMS) values of the time series of accelerations for periods of 30 min during 1 month, as shown in Figure 34. It can be observed that sensors present a proper functioning since it is possible to note consecutive peaks of acceleration on business days and a decrease in these values on the weekends and holidays (3 June and 10 June). Differences between vertical (green) and lateral (blue) measurements are observable as well.

After that, the application of the SSI-Cov method for a specific record of 30 min belonging to that month allowed us to obtain a stabilization diagram. To better distinguish the results obtained in lateral and vertical directions, this diagram was generated for both cases, as shown on Figure 35 and Figure 36. Similar natural frequencies can be found in both diagrams, indicating the occurrence of vibration modes with components in the two directions.

A procedure based in Cluster Analysis [27], proposed and described in [28], was used to perform automated operational modal analysis and continuously track the modal properties of the first 14 vibration modes of the structure. This processing of the acceleration signals was performed continuously for every 30 min long group of time-series, during June 2021, allowing to evaluate the evolution of the natural frequencies for each vibration mode in this period, which are represented by different color dots in Figure 37. Given the high number of modes, the analysis was focused on the frequency band 0–4 Hz. When looking in detail the obtained frequencies, it is possible to detect small variations over time, that are likely to be due to temperature and traffic influence.

This analysis proceeded to the damping results (see Figure 38), permitting the identification of the significantly increased damping observed during some business days periods. This increase may be attributed to the higher traffic intensity during some periods, which is a phenomenon detected in other case studies, such as with the Infante D. Henrique bridge [29]. However, the sharp increase in damping of the 1st vertical vibration mode (dark orange colored), which has a longitudinal component, may be amplified by the friction present in the expansion joints. This is an important issue to be investigated that will be addressed in future work.

The average frequencies and damping ratios obtained by analysing the results from June 2021 in the range of 0–4 Hz are presented in Table 4, where frequencies are compared to the experimental results obtained from the ambient vibration tests (AVTs) using the Peak-Picking method [30].

It can be observed that the natural frequencies obtained from the ambient vibration tests are also identified in the continuous monitoring results obtained using the SSI-Cov method. However, there are additional frequencies found when using SSI-Cov (modes 4, 5, 8, 11, 13 and 14) that were not detected in the ambient tests, whose origin has yet to be studied. This may arise from the application of different identification methods to the signals but also from the complexity of the shape of these vibrations modes, which were identified with the SSI-Cov method, but generally present lower identification rates than those already identified with the AVTs.

## 6. Conclusions

This paper describes the recurring structural monitoring of a large-pan arch bridge using customized sensors, namely, accelerometers, temperature sensors and displacement sensors. It was noted that these types of sensors have several advantages over traditional solutions, particularly related to their direct installation and low costs, greater flexibility, high autonomy, lower maintenance and the fact that they facilitated equipment replacement or repair.

Concerning the temperature measurements carried out by these devices, it was possible to observe the different profiles obtained in the sections of the deck and in the arch of the bridge, either in terms of temperature amplitudes or phase shifts. The sensors installed on the arch recorded much higher temperature variations and sharper peaks than the devices installed on the deck. Moreover, temperature curves measured in relation to the deck are delayed approximately 5.3 h relative to the temperature recorded for the arch. This is due to the higher exposure of the arch sections to sunlight incidence when compared to the inferior side of the deck where the deck sensors are installed.

One of the important contributions of the measurements of displacements in the expansion joints was the establishment of a correlation law between temperature and displacements, which allowed for the estimation of the movements of the expansion joints with good accuracy when only analysing the temperature measured on the deck. In addition, the continuous measurement of displacement in the expansion joints allowed for the identification of a dynamic component that is related to the intensity of traffic loads, which will allow for the future estimation of the traffic loads acting on the structure. The observation of this phenomenon is an innovative aspect found in this investigation, which, as far as the authors are aware, is the first time that this has been referred to in scientific articles.

Regarding the measurements performed with accelerometers using OMA tools, it was possible to identify a more complete set of natural frequencies and mode shapes beyond those that were previously obtained with ambient vibration tests.

## Figures and Tables

**Figure 1 sensors-23-05971-f001:**
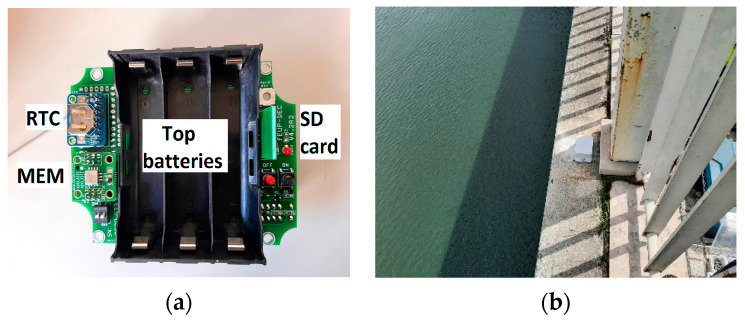
Accelerometer modules: (**a**) view inside the module; (**b**) view on the bridge.

**Figure 2 sensors-23-05971-f002:**
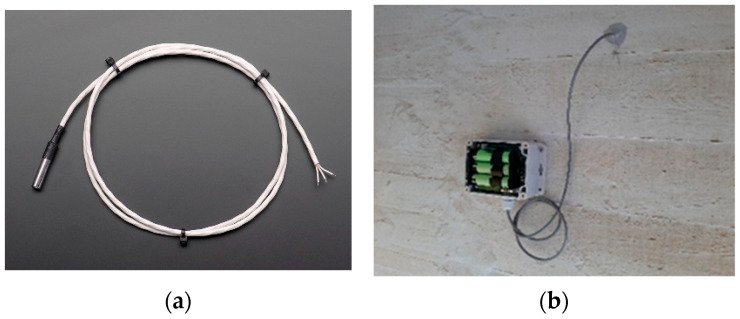
Temperature sensors: (**a**) view inside the module; (**b**) view on the bridge.

**Figure 3 sensors-23-05971-f003:**
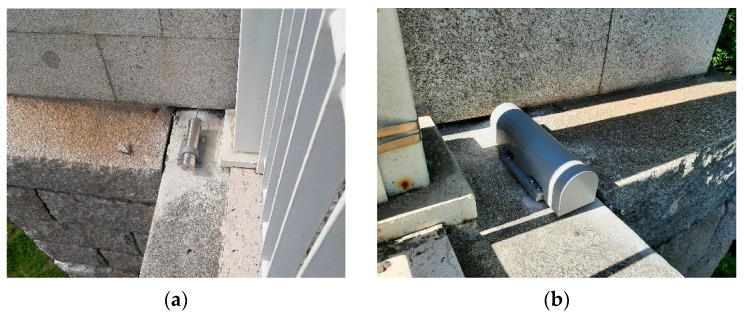
Displacement sensor at expansion joint: (**a**) view of the cylinder enclosure; (**b**) external protection.

**Figure 4 sensors-23-05971-f004:**
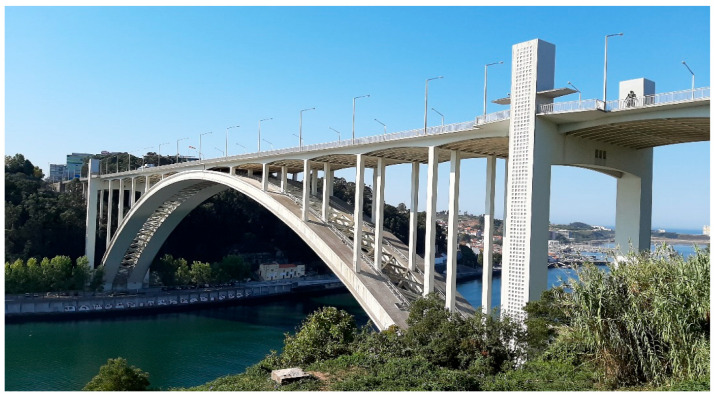
Arrábida bridge, Porto—Portugal.

**Figure 5 sensors-23-05971-f005:**
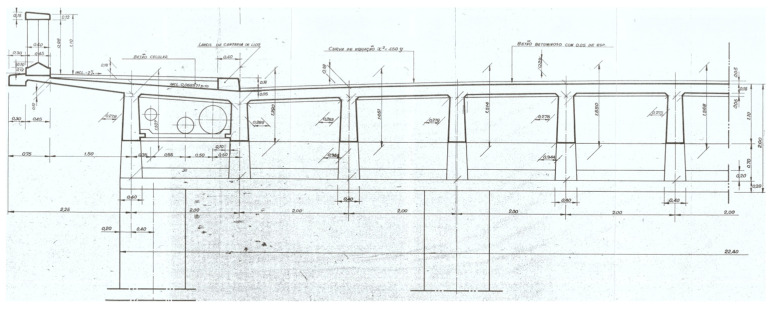
Transversal section of the deck.

**Figure 6 sensors-23-05971-f006:**
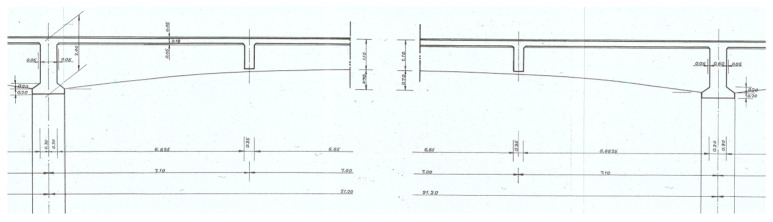
Longitudinal beams with a 21.2 m span.

**Figure 7 sensors-23-05971-f007:**
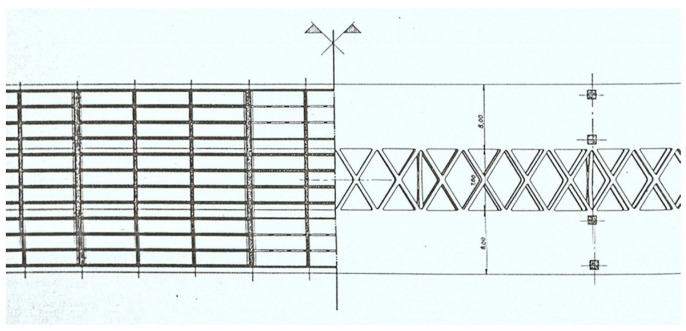
Plan view of the beams and twin arches.

**Figure 8 sensors-23-05971-f008:**
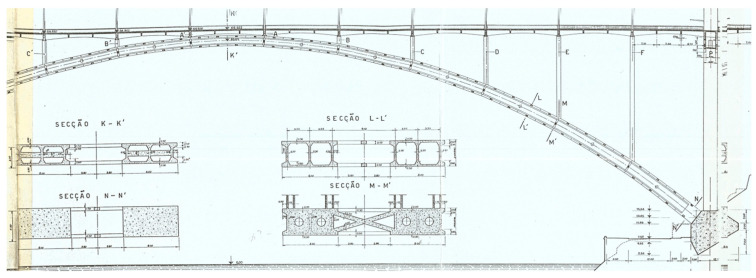
Different cross-sections of the arches and respective alignments.

**Figure 9 sensors-23-05971-f009:**
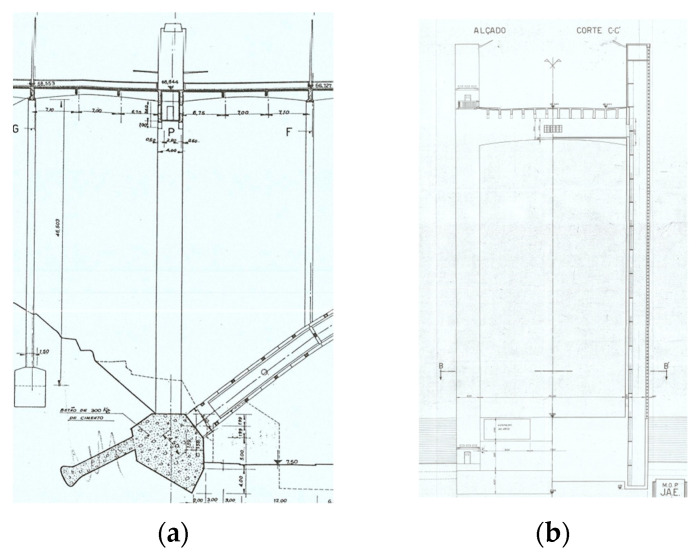
Pilasters of the bridge: (**a**) longitudinal view; (**b**) transversal view.

**Figure 10 sensors-23-05971-f010:**
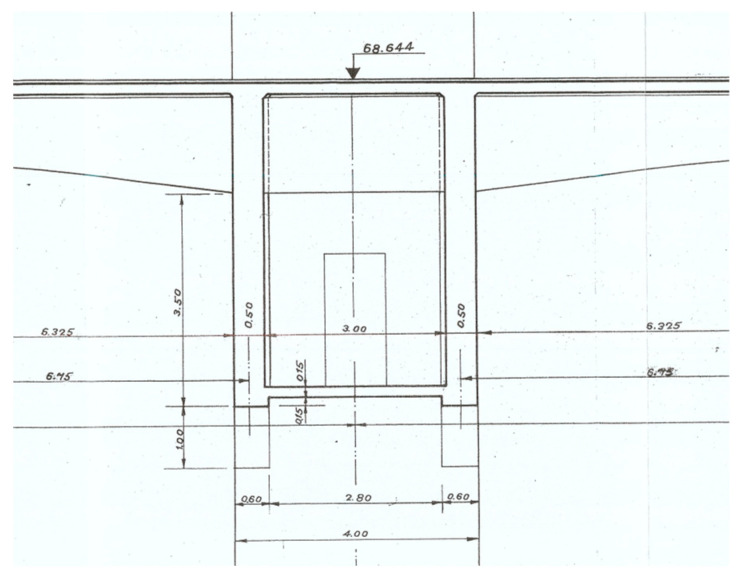
Cross-section of the beams that connect the pilasters.

**Figure 11 sensors-23-05971-f011:**
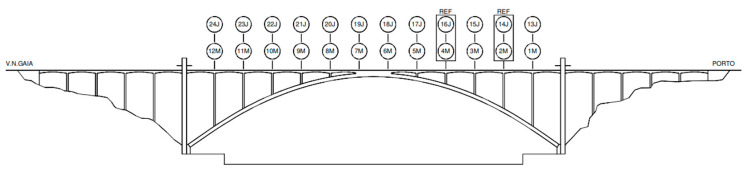
Location of the measurement points and fixed stations.

**Figure 12 sensors-23-05971-f012:**
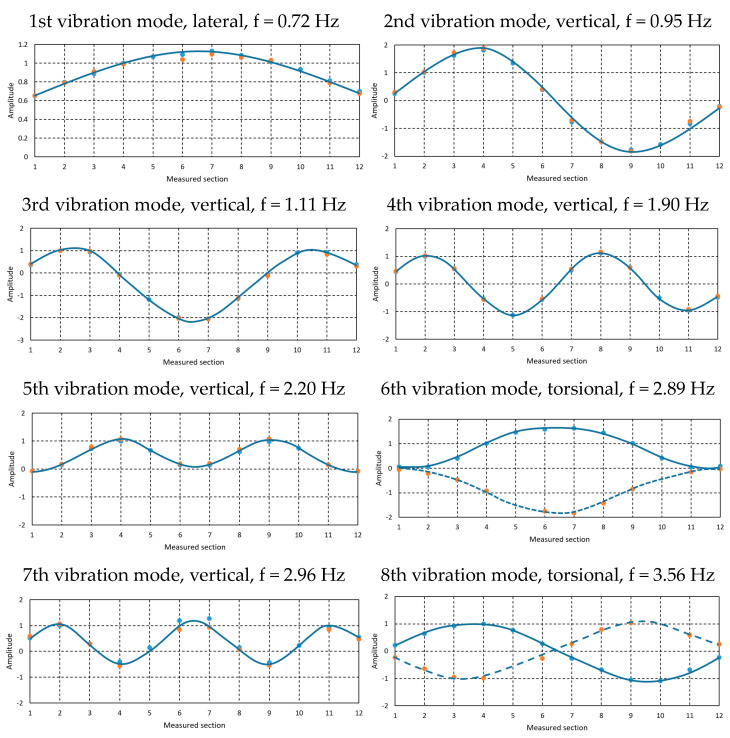
Natural frequencies and vibration modes obtained in the ambient vibration tests.

**Figure 13 sensors-23-05971-f013:**
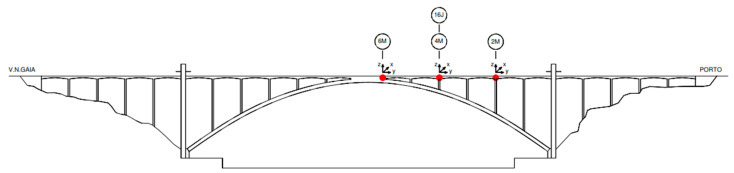
Location of the instrumented sections of the bridge.

**Figure 14 sensors-23-05971-f014:**
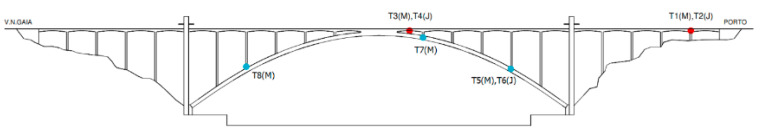
Location of the sections instrumented with temperature sensors.

**Figure 15 sensors-23-05971-f015:**
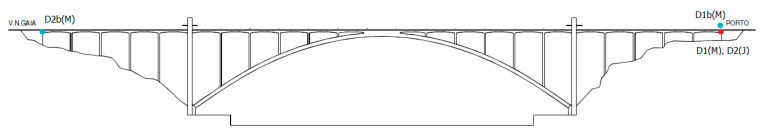
Location of the sections instrumented with displacement sensors.

**Figure 16 sensors-23-05971-f016:**
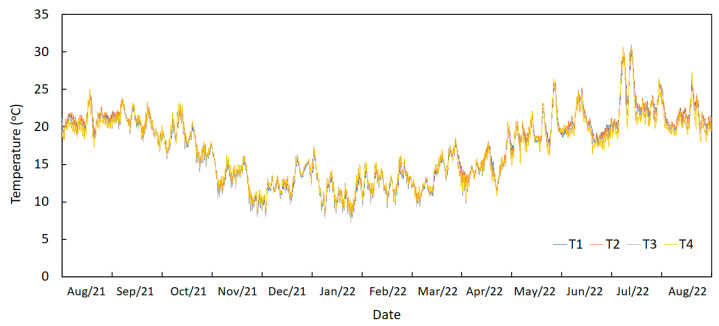
Temperature records from August 2021 to August 2021 measured by sensors T1, T2, T3 and T4.

**Figure 17 sensors-23-05971-f017:**
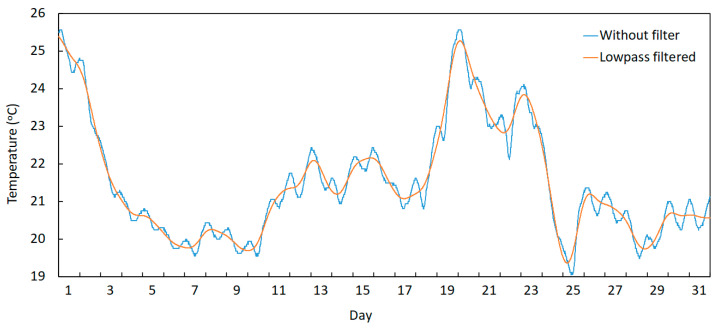
Temperature variation of sensor T1 during August 022: obtaining the trend component by applying a lowpass filter.

**Figure 18 sensors-23-05971-f018:**
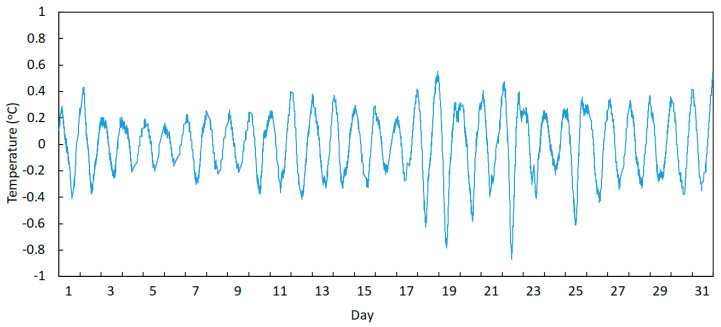
Temperature variation of sensor T1 during August 2022: obtaining the daily component by applying a highpass filter.

**Figure 19 sensors-23-05971-f019:**
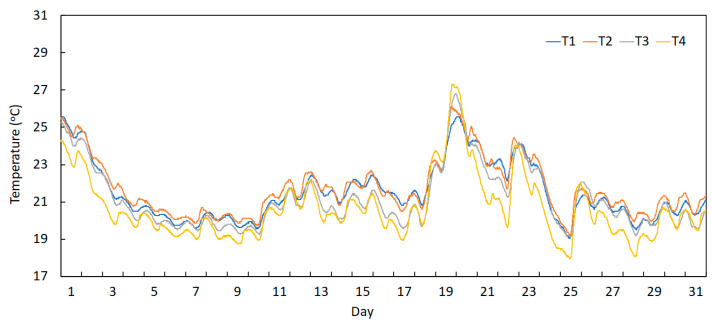
Temperature records during August 2022 measured using the deck sensors (T1 to T4).

**Figure 20 sensors-23-05971-f020:**
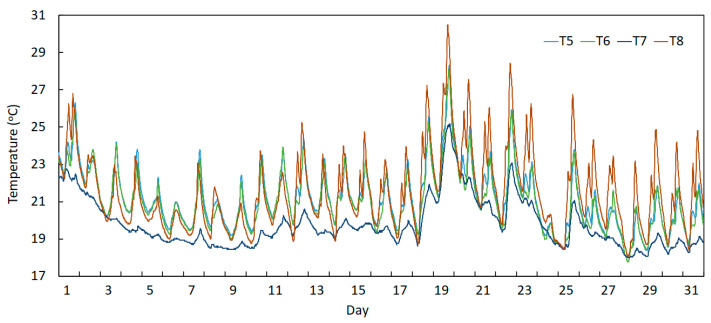
Temperature records during August 2022 measured using the arch sensors (T5 to T8).

**Figure 21 sensors-23-05971-f021:**
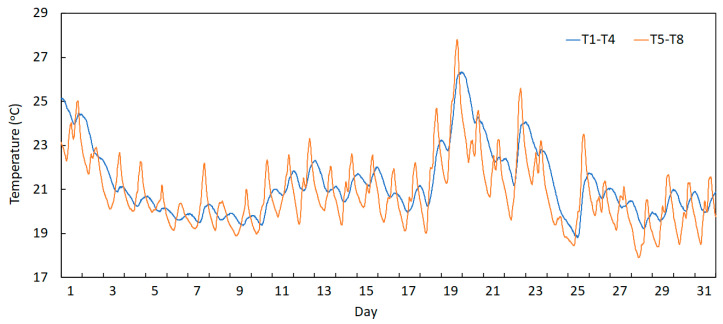
Average temperature variation in August 2022 of the two groups of sensors: deck sensors (T1 to T4) and arch sensors (T5 to T8).

**Figure 22 sensors-23-05971-f022:**
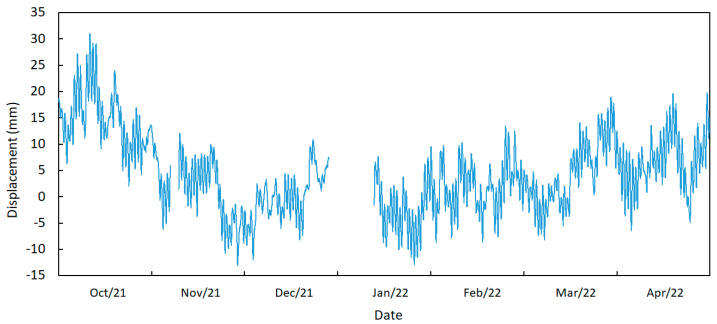
Displacement variations in the north joint during September 2021 to May 2022 (average of sensors D1 m and D2-J).

**Figure 23 sensors-23-05971-f023:**
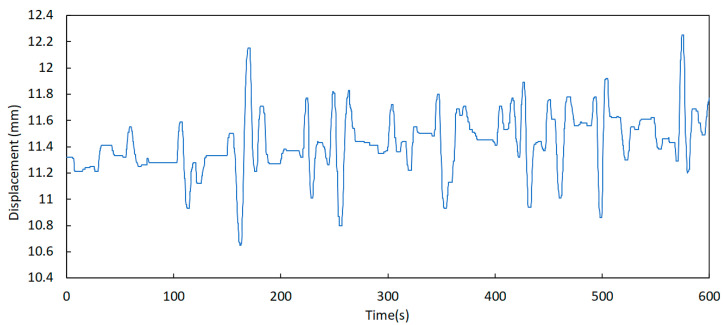
Displacement variation in sensor D1 on 22 October 2021 at 14 h 20 min (example of the dynamic component).

**Figure 24 sensors-23-05971-f024:**
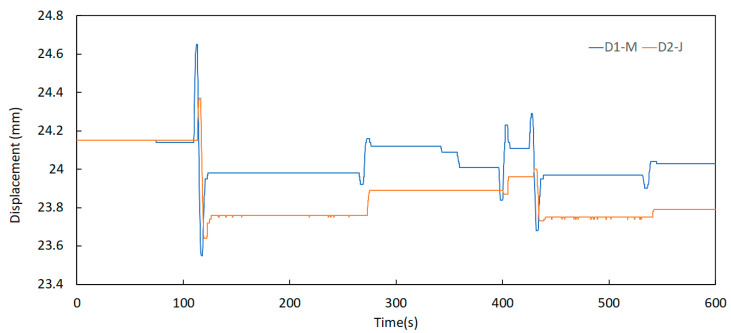
Displacement signals of the north joint sensors D1—M and D2—J on 11 October 2021 at 00 h 00 min.

**Figure 25 sensors-23-05971-f025:**
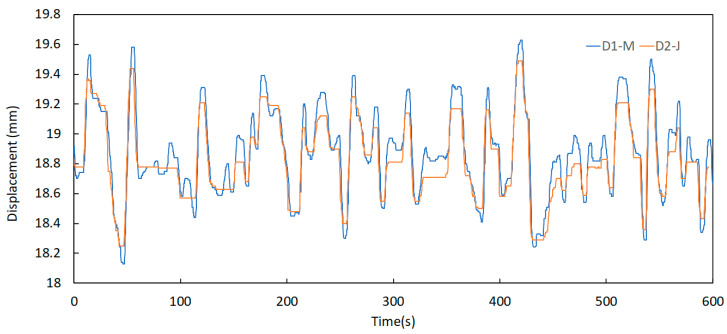
Displacement variation of the north joint sensors D1—M and D2—J on 11 October 2021 at 08 h 00 min.

**Figure 26 sensors-23-05971-f026:**
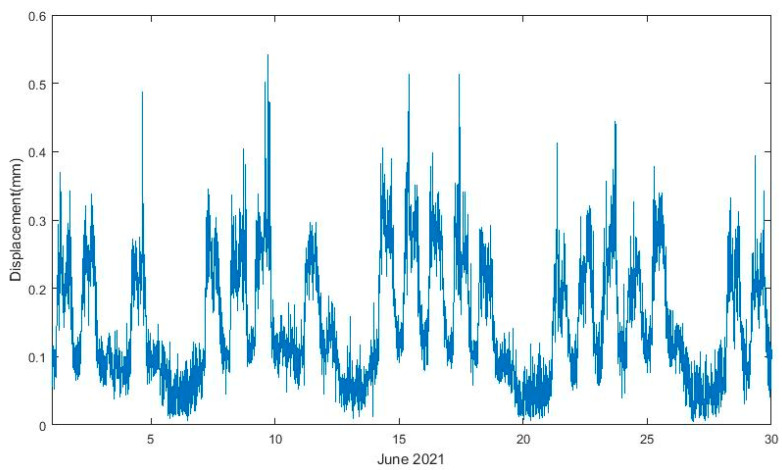
RMS values of displacements measured by sensor D1 m during June 2021.

**Figure 27 sensors-23-05971-f027:**
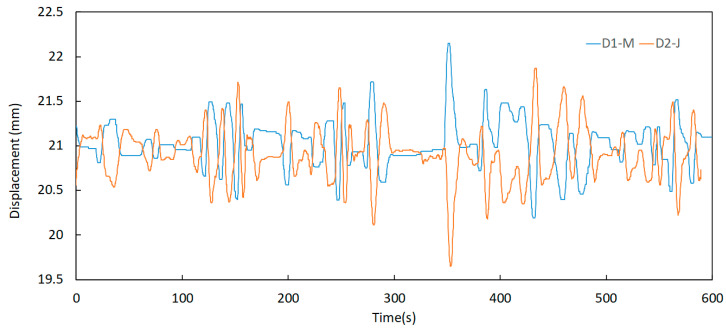
Displacement variation of the north and south joint sensors D1b—M and D2b—M on 12 September 2022 at 08 h 00 min.

**Figure 28 sensors-23-05971-f028:**
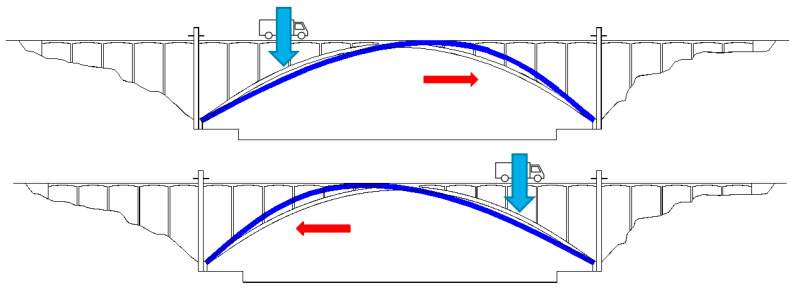
Sketch of the deformation of the arch when submitted to a vertical loading in each half-span.

**Figure 29 sensors-23-05971-f029:**
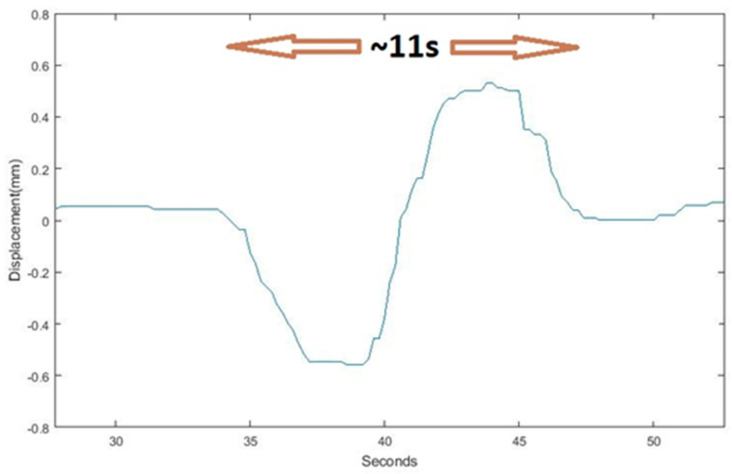
Example of a sinusoidal motion of the arch in the longitudinal direction.

**Figure 30 sensors-23-05971-f030:**
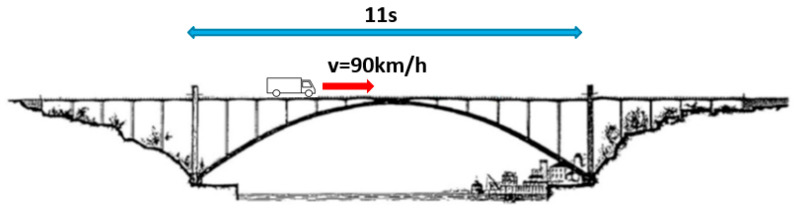
Estimation of the velocity of the traffic loads.

**Figure 31 sensors-23-05971-f031:**
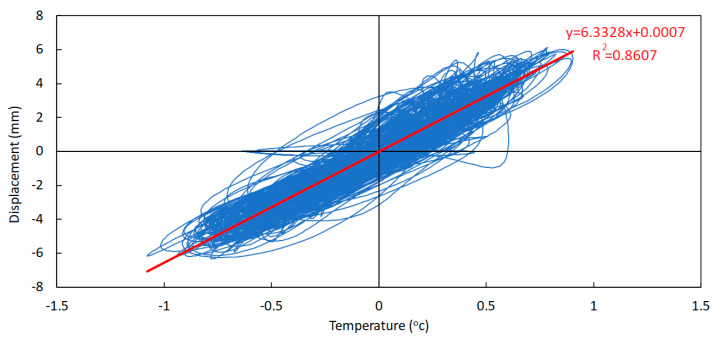
Temperature-versus-displacement chart—daily components (September 2021 to May 2022).

**Figure 32 sensors-23-05971-f032:**
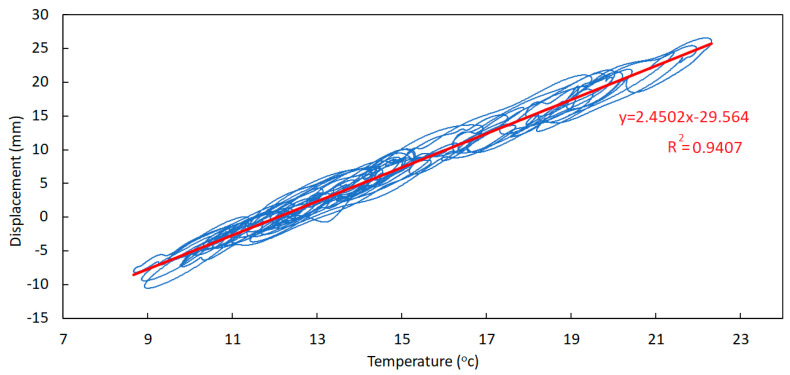
Temperature-versus-displacement chart—trend components (September 2021 to May 2022).

**Figure 33 sensors-23-05971-f033:**
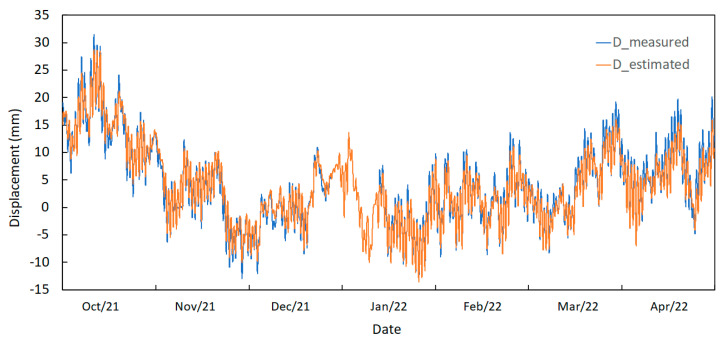
Displacement curves (obtained and estimated values during September 2021 to May 2022).

**Figure 34 sensors-23-05971-f034:**
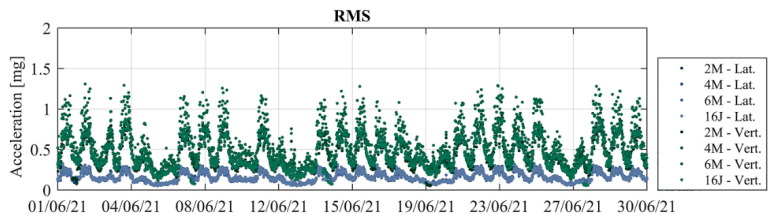
RMS values of accelerations measured during June 2021.

**Figure 35 sensors-23-05971-f035:**
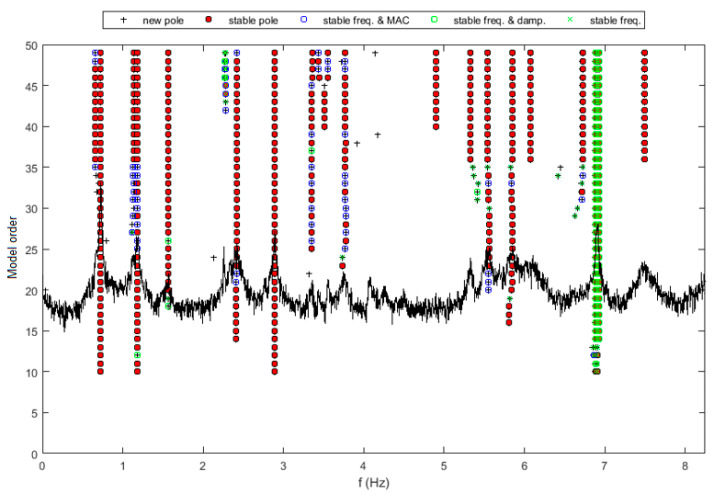
Stabilization diagram obtained using SSI-Cov methodology for lateral accelerations.

**Figure 36 sensors-23-05971-f036:**
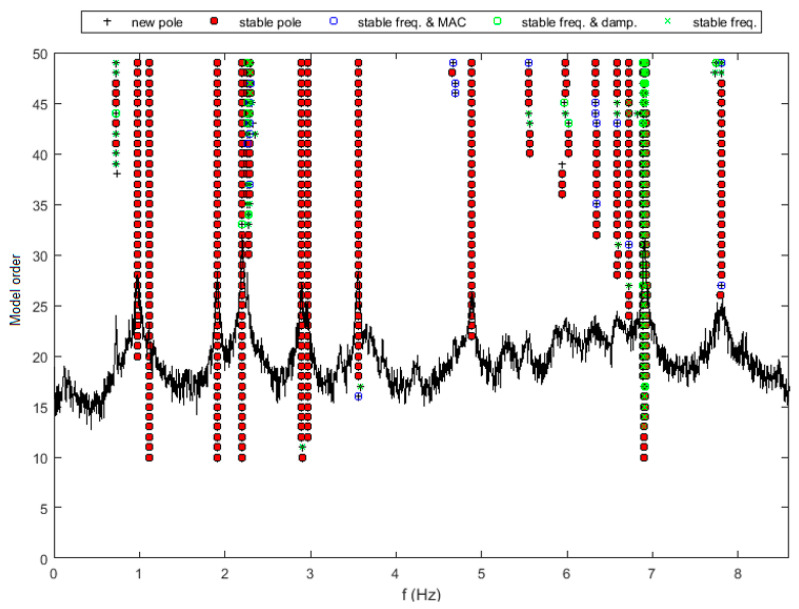
Stabilization diagram obtained using SSI-Cov methodology for vertical accelerations.

**Figure 37 sensors-23-05971-f037:**
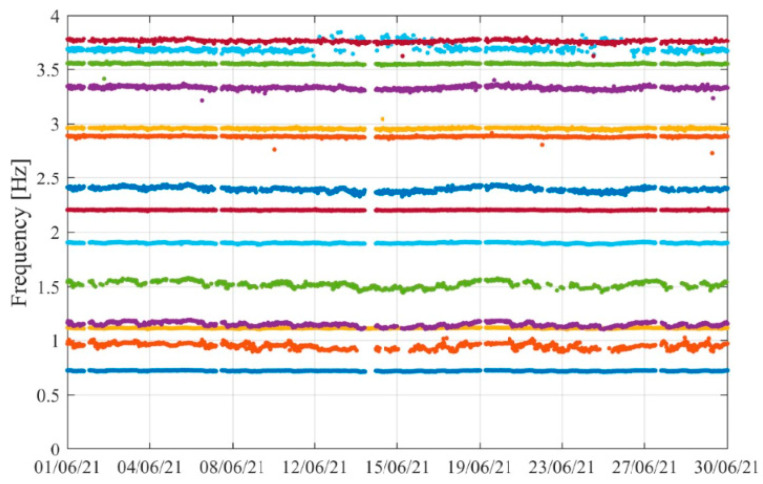
Frequency evolution along time (results obtained during June 2021).

**Figure 38 sensors-23-05971-f038:**
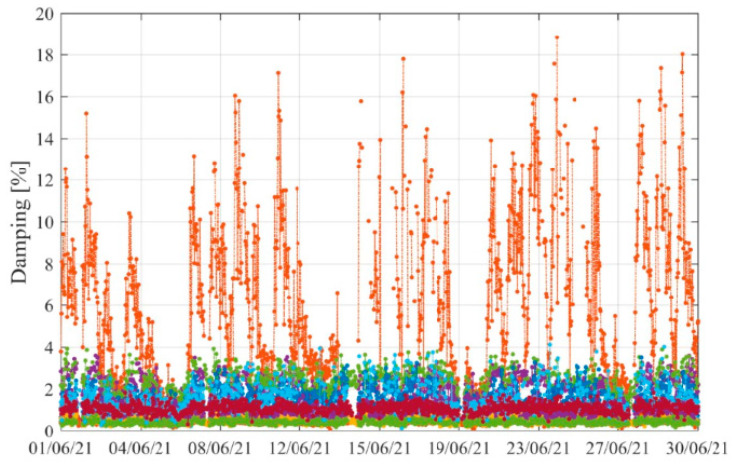
Damping evolution along time (results obtained during June 2021).

**Table 1 sensors-23-05971-t001:** Vibration modes obtained in the ambient vibration tests.

Vibration Mode	Direction	Natural Frequency
1	Lateral	0.72 Hz
2	Vertical	0.95 Hz
3	Vertical	1.11 Hz
4	Vertical	1.90 Hz
5	Vertical	2.20 Hz
6	Torsional	2.89 Hz
7	Vertical	2.96 Hz
8	Torsional	3.56 Hz

**Table 2 sensors-23-05971-t002:** Statistical values in (°C) of the measured annual temperature.

Sensor	T1	T2	T3	T4
Maximum	29.75	30.19	31.00	30.62
Minimum	8.06	8.00	7.25	8.00
Average	17.10	17.34	17.00	17.03

**Table 3 sensors-23-05971-t003:** Statistical values of the displacement in the north expansion joint.

Displacement (mm)	Average of D1 m and D2-J
Maximum	31.043
Minimum	−13.058
Peak to peak	44.101

**Table 4 sensors-23-05971-t004:** Average frequencies and damping ratios obtained using the SSI-Cov method.

Mode	AVT Frequency (Hz)	Average Frequency (Hz)	Average Damping (%)	Mode Shape	Identification Success Rate (%)
1	0.72	0.72	0.84	Lateral	95.1
2	0.95	0.95	6.16	Vertical/(slightly) Longitudinal	82.2
3	1.11	1.11	0.45	Vertical	98.4
4	-	1.15	2.18	Lateral/Vertical Torsion	74.9
5	-	1.52	2.40	Lateral/Longitudinal Torsion	52.3
6	1.90	1.90	0.58	Vertical	95.0
7	2.20	2.20	0.54	Vertical	96.7
8	-	2.39	1.58	Lateral/Vertical Torsion/Longitudinal Torsion	80.0
9	2.89	2.88	0.46	Vertical	96.5
10	2.96	2.96	0.54	Vertical	94.9
11	-	3.33	1.07	Lateral/Vertical Torsion	93.7
12	3.56	3.55	0.43	Vertical Torsion	94.9
13	-	3.69	1.62	Vertical Torsion/Longitudinal Torsion	51.7
14	-	3.76	1.10	Lateral/Vertical Torsion	85.5

## Data Availability

The data presented in this study are available on request from the corresponding author.

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
