# Peer review of "Structural Monitoring of a Large-Span Arch Bridge Using Customized Sensors"

_sensors, 2023, doi:10.3390/s23135971_

Round 1

Reviewer 1 Report

In this paper, the displacement of bridge expansion joint is estimated by temperature sensor system, which is an interesting and full of engineering work. However, this article still needs some modifications before it can be published.

1. There are few references cited in this paper, so it is suggested to investigate and summarize the relevant research.

2. It is suggested to add a flow chart to summarize the work of the whole paper.

3. There is an error in the reference of 520 lines on page 21.

4. Can the wave velocity measurement method provided in Figure 28 be applied in the rush hour?

5. Are relevant verification experiments carried out to verify the fitting formula according to the actual displacement?

Author Response

1. There are few references cited in this paper, so it is suggested to investigate and summarize the relevant research.
R: The literature review was expanded, adding 1 more page to the article. The references list was updated with a significant more references.  
2. It is suggested to add a flow chart to summarize the work of the whole paper.
R: The reviewer's suggestion is helpful. However, as this research work is ongoing and not yet finished, the inclusion of a complete flow chart will make more sense at a later stage, whose work will be published in a subsequent article including more advances in this investigation.
3. There is an error in the reference of 520 lines on page 21.
R: Correction done.
4. Can the wave velocity measurement method provided in Figure 28 be applied in the rush hour?
R: Yes. We are using the wave velocity to make an extensive study of the traffic characteristics over long periods. Not only on terms of vehicles velocity, but also in terms of loading. This work will be published in a subsequent article.     
5. Are relevant verification experiments carried out to verify the fitting formula according to the actual displacement?
R: Yes. We think that although the correlation between these two variables may be obtained analytically by considering the properties of the construction materials, it is useful and more reliable to perform an experimental calibration.

Reviewer 2 Report

This study presents structural health monitoring technology of a long-span arch bridge. The vibration behavior and static mechanical behavior have been investigated. The SHM data provide a theoretical basis for bridge safety evaluation. This study deserves investigation, and the manuscript is well-organized. The manuscript can be published after minor revisions. Please consider the following comments:

1. The introduction is insufficient. There is a great amount of SHM references. A state-of-the-art review on the bridge SHM development should be added.

2. The bridge figures related to Fig. 5 to Fig. 10 are difficult to read since the label of the structure cannot be seen clearly.

3. Table 1, the measured frequency can be compared with numerical results estimated from FEM.

4. This study seems like a technical report. More discussions on theoretical research should be provided.

5. The references are relatively insufficient. 

The English writing is good.

Author Response

1. The introduction is insufficient. There is a great amount of SHM references. A state-of-the-art review on the bridge SHM development should be added.
R: The literature review was expanded, adding 1 more page to the article. The references list was updated with a significant more references.  
2. The bridge figures related to Fig. 5 to Fig. 10 are difficult to read since the label of the structure cannot be seen clearly.
R: These Figures were obtained from digitized drawings of the original project. We don't think it is possible to get better quality. At the time of final editing of the paper, if this is a problem, we will try to get a better scan of the images if really needed.
3. Table 1, the measured frequency can be compared with numerical results estimated from FEM.
R: This article is focused on the analysis and interpretation of experimental data obtained with customized sensors. Although a calibrated numerical model exists, the inclusion of numeric studies is out of the scope this paper. 
4. This study seems like a technical report. More discussions on theoretical research should be provided.
R: As just mentioned, the purpose of this paper is to highlight the potential of an in-situ monitoring system. The work of theoretical and numerical nature is being developed and will be published in the future in an article with other objectives.
5. The references are relatively insufficient.
R: As mentioned, the references list was updated with a significant more references.  

Reviewer 3 Report

A) General remarks

The research presents in this paper a very interesting topic, as well as results that are of wider significance when it comes to the ground motion effects examination on civil engineering structures. The paper is concise and clear. The literature in the paper is adequately cited but very basic, however, some comments on the choice and significance of cited sources will be articulated in the points below.
1.    The complete affiliation must be given according to the journal template
2.    In the case of literature, it must be pointed out that this aspect is one of the biggest problems of the article. 8 positions suggest no proper evaluation of the current state of the art. As the article is in the field very well examined, this must be improved.
3.    The biggest problem of the article is the description of the novelty of the research without it the paper presenting engineering tools is questionable as an important scientific topic. Please write clearly what the authors input to the field. This information is not included also in the conclusions.
4.     It is suggested to make small changes to the abstract. The role of the abstract is to give a basic overview of the paper. There is no mention of the novelty of the paper and from just the abstract it seems a bearly paper showing some engineering approach to the problem.
5.    The introduction is basic and rushed. E.g:
a.    No proper introduction to structural monitoring, seismic monitoring of civil engineering structures etc.
b.    Authors go straight to OMA but also classical EMA is being used.
c.    No information on measuring techniques with a division on contact (accelerometers, geophones, displacement sensors) and none- contact optical methods like digital image correlation (e.g https://doi.org/10.1155/2021/6694790) or 3D laser vibrometry used in both lightweight and large-span structures like bridges (e.g doi:10.18429/JACoW-IPAC2018-WEPMF079).
d.    As authors are developing their own sensors it is worth evaluating measuring techniques as in point “c” but also focus a paragraph on the use of specific accelerometers for structural health monitoring (https://doi.org/10.3390/jsan7030030), evaluation of structure and machines (DOI: 10.1051/e3sconf/201910000080), improving simulations etc.
e.    Similarly “d” displacement sensors and their usage require an introduction.
f.     Please improve and give a proper evaluation of methods.
g.    The last paragraph of the introduction is reserved for aim, scope and novelty. This is very minimalistic in the article. Again, with no novelty component. The statement provided suggests this is a more technical report than a scientific paper.

6.    Chapter 2- was the custom-made accelerometer tested against a commercial one? What about the synchronization- is it reliable enough for the purpose of such measurements? Best to give some references proving that this is the correct approach.
7.    Chapter 3 is ok
8.    The results- some of them are presented in figures that are bearly visible. Especially the ones with the time stamps on x-axis. Suggest to stamp every few days not every day so the graphs are clear (like graph 34).
9.    The conclusions are good.

B) Item remarks
Fig 5-10. Are very low quality. If possible please improve.
Fig. 29 is very low quality. If possible please improve.

C) Conclusions:
The biggest problem of the article is also the clear presentation of the novelty of the research topic, although this is much better presented in the conclusions.
The part that requires major changes is the introduction which does not offer a proper state-of-the-art presentation. Some changes/answers are also required in the case of custom-made accelerometers and communication.
However, the reviewer acknowledges a huge amount of work from the authors, and a good presentation of results, and is willing to mark the paper for minor revisions only if all elements mentioned in this revision are corrected thoroughly.

The article is interesting and well-written.

The biggest problem of the article is also the clear presentation of the novelty of the research topic, although this is much better presented in the conclusions.
The part that requires major changes is the introduction which does not offer a proper state-of-the-art presentation. With bearly 8 references the real state-of-the-art was not performed.

Some changes/answers are also required in the case of custom-made accelerometers and communication.

Author Response

1.    The complete affiliation must be given according to the journal template
R: Correction done.
2.    In the case of literature, it must be pointed out that this aspect is one of the biggest problems of the article. 8 positions suggest no proper evaluation of the current state of the art. As the article is in the field very well examined, this must be improved.
R: The literature review was expanded, adding 1 more page to the article. The references list was updated with a significant more references.  
3.    The biggest problem of the article is the description of the novelty of the research without it the paper presenting engineering tools is questionable as an important scientific topic. Please write clearly what the authors input to the field. This information is not included also in the conclusions.
R: The main contribution of this work is to demonstrate the ease with which a structure can be instrumented using an affordable solution, but which is at the same time precise and robust. This monitoring system includes the development of a small displacement sensor at expansion joints that works with batteries with an autonomy of more than 1 year and has a resolution of 0.01mm, which is remarkable. In addition, the observation of a dynamic component in the displacement of the expansion joints resulting from traffic loads is an innovating finding, which, as far as the authors are aware, is the first time that is referred in scientific articles. These contributions to the state of the art were reinforced and clarified in different parts of the article, including the abstract, the body of the document and the conclusions.
4.     It is suggested to make small changes to the abstract. The role of the abstract is to give a basic overview of the paper. There is no mention of the novelty of the paper and from just the abstract it seems a bearly paper showing some engineering approach to the problem.
R: In the authors' opinion, the updated abstract is faithful to the content of the article, describing all the work carried out to date involving this bridge, and referring to the main results and contributions.
5.    The introduction is basic and rushed. E.g:
a.    No proper introduction to structural monitoring, seismic monitoring of civil engineering structures etc.
R: The introduction was updated with more details about structural monitoring. Seismic monitoring doesn’t fit in the scope of this paper; therefore, it was not developed. 
b.    Authors go straight to OMA but also classical EMA is being used.
R: The framework of EMA techniques was included in the introduction section. 
c.    No information on measuring techniques with a division on contact (accelerometers, geophones, displacement sensors) and none- contact optical methods like digital image correlation (e.g https://doi.org/10.1155/2021/6694790) or 3D laser vibrometry used in both lightweight and large-span structures like bridges (e.g doi:10.18429/JACoW-IPAC2018-WEPMF079).
R: There is a huge variety of measurement techniques in Structural Monitoring. Mentioning them here would be tedious, and would divert the reader's attention to other topics that are not covered in detail in this article. At the same time, referring only some techniques would be vague and imprecise. For this reason, we emphasized these techniques in the description of the developed sensors.
d.    As authors are developing their own sensors it is worth evaluating measuring techniques as in point “c” but also focus a paragraph on the use of specific accelerometers for structural health monitoring (https://doi.org/10.3390/jsan7030030), evaluation of structure and machines (DOI: 10.1051/e3sconf/201910000080), improving simulations etc.
R: The first suggested reference is pertinent in the context of this work, which was introduced and included in the body of the document.
e.    Similarly “d” displacement sensors and their usage require an introduction.
R: The functioning and the usage of the displacement sensors are described in section 2.3.
f.     Please improve and give a proper evaluation of methods.
R: Answered before.
g.    The last paragraph of the introduction is reserved for aim, scope and novelty. This is very minimalistic in the article. Again, with no novelty component. The statement provided suggests this is a more technical report than a scientific paper.
R: This part of the introduction was modified accordingly.
6.    Chapter 2- was the custom-made accelerometer tested against a commercial one? What about the synchronization- is it reliable enough for the purpose of such measurements? Best to give some references proving that this is the correct approach.
R: In fact, the accelerometer “module” is not commercial, but the digital accelerometer itself is commercial and well-known one. This device was tested and used in a previous work involving scaffolding systems for bridge construction. A reference was included where this accelerometer module was compared to a commercial system.   
7.    Chapter 3 is ok
8.    The results- some of them are presented in figures that are bearly visible. Especially the ones with the time stamps on x-axis. Suggest to stamp every few days not every day so the graphs are clear (like graph 34).
R: All graphs with stamps on x-axis were updated to a better appearance and clarity. 
9.    The conclusions are good.
B) Item remarks
Fig 5-10. Are very low quality. If possible please improve.
R: These Figures were obtained from digitized drawings of the original project. We don't think it is possible to get better quality. At the time of final editing of the paper, if this is a problem, we will try to get a better scan of the images if really needed.
Fig. 29 is very low quality. If possible please improve.
R: The original figure has this quality. At the time of final editing, if needed, we produce a new Figure.  

Reviewer 4 Report

The paper

“Structural Monitoring of a Large-Span Arch Bridge Recurring to Customized Sensors”,

By

Ietka et al.,

Presents a case study of customised sensors for bridge monitoring. In particular, these newly-developed devices are applied to a reinforced concrete arch bridge, the Arrábida Bridge.

The paper is very interesting for researchers and practitioners in civil engineering, especially for the ones more focused on dynamic monitoring, OMA, and output-only System Identification; all topics for which researchers at FEUP are well-known and highly appreciated.

Firstly, the paper describes the proposed sensors and presents the case study. Then, it introduces the preliminary ambient vibration tests used to identify the natural frequencies and the vibration modes of the deck. The paper goes on to describe measurement campaigns that involved different types of sensors, including accelerometers, temperature sensors, and displacement sensors.

The paper is definitely worth attention but it presents some conceptual and editorial issues. Thus, before being fully accepted, the following major and minor remarks should be addressed.

Major (conceptual) remarks:

1.      Figure 1.a: as the main point of this paper concerns the newly proposed sensors, it could be useful to include more technical details. E.g., text arrows could be used to point and highlight the several electronic components (correctly described in Sec 2.1)

2.      Regarding the results of the AV tests. The first mode (0.72 Hz) is claimed to be a lateral mode. However, it clearly is (also?) the first flexural one in the vertical direction (assuming that ‘amplitude’ means vertical amplitude). Plus, it is not clear why all the other mode shapes have been normalised between [-2,+2] but only this first one spans from 0 to 1.2

3.      Figure 22 (and elsewhere). There are some gaps in the recording, which is unfortunately understandable. It may seem pedantic but, rather than reporting these gaps with a constant value (which is conceptually incorrect), it would be better to report no data at these timesteps.

4.      Figures 31 and 32. Several data points seem to suggest displacements at zero degrees. Is this true or is it due to sensor faults (or something else)?

5.      Figure 21: the results of these analyses showed that the temperature measured at the deck sections presented different amplitudes and phase shifts when compared to the temperature measured at the arch sections. Indeed, the deck temperature is clearly lagging behind the arch temperature. However, by comparing it with the intraday variation of the natural frequencies (if available), which of the two most closely follow them? Related to this aspect: would the Authors suggest placing temperature sensors on the deck or at the arches, if they had to choose one of the two subsets?

6.      The paper proves that by using temperature measurements, it is possible to estimate the displacements in the expansion joints of the bridge accurately. Again, this was assessed by using sensors T1 to T4 (located on the deck). Does it mean that temperature sensors located on the deck are more predictive than temperature sensors located on the arches, for what concerns the displacements in the expansion joints?

7.      In the Introduction, the state-of-the-art review for bridge output-only monitoring can be extended. There are many recent studies, which also makes use of the same SSI-COV technique used here, both for masonry arch (https://doi.org/10.1002/stc.3028) or R.C. road bridges (https://doi.org/10.1016/j.measurement.2023.112451). These could be reported to provide some further context.

Editorial and minor remarks:

1.      Line 385, page 14; line 520, page 21: “Figure 385 23Error! Reference source not found.”

2.      Since all authors belong to the same institution (FEUP), is not necessary to use different numbers in the Authors’ List.

3.      Figure 2.b: by looking at it, it seems more like the accelerometer module of Figure 1.a than the sensor of Figure 2.a. Is it correct? From the text, it seems to understand that indeed the same external case was used. In that case, it may be useful to also portray the sensors of Figure 2.a as they appear inserted inside the polycarbonate box.

4.      Figure 18 and elsewhere: please add the legend

5.      There are issues from Figure 16 to 22 (also Figure 33 and elsewhere). Specifically, it should be 2021 and 2022, not 0021 and 0022.

7.      Overall, the content of the paper is very interesting and the data are clearly exposed but the quality of the Figure is a bit low. Some are grainy (e.g. Figures, 26, 37, 38, etc).

8.      Page 18: it is not clear why ‘versus’ should be in italics.

9.      Page 19 and elsewhere: all equations should be numbered.

10.   Figures 5 to 10. This is a mere (and non-binding) stylistic suggestion but, if available, AutoCAD drawings could be better than the scanned original plans. 

The English of the paper is overall good but it may use some grammar checking. There are a few mistakes and typos throughout the manuscript (e.g. page 18, line 456, ‘de’ instead of ‘the’).

Author Response

1. Figure 1.a: as the main point of this paper concerns the newly proposed sensors, it could be useful to include more technical details. E.g., text arrows could be used to point and highlight the several electronic components (correctly described in Sec 2.1)
R: Correction done.
2. Regarding the results of the AV tests. The first mode (0.72 Hz) is claimed to be a lateral mode. However, it clearly is (also?) the first flexural one in the vertical direction (assuming that ‘amplitude’ means vertical amplitude). Plus, it is not clear why all the other mode shapes have been normalised between [-2,+2] but only this first one spans from 0 to 1.2
R: Definitely 0.72 Hz is a lateral mode and the others are vertical or torsion. Another earlier modal identification confirms this.
Regarding normalization, depending on the reference station adopted, the amplitudes may vary. It would be better to normalize to 1, but it would give a lot of graphical work. So, if possible, we like to maintain these Figures as they are.
3. Figure 22 (and elsewhere). There are some gaps in the recording, which is unfortunately understandable. It may seem pedantic but, rather than reporting these gaps with a constant value (which is conceptually incorrect), it would be better to report no data at these timesteps.
R: Correction done.
4. Figures 31 and 32. Several data points seem to suggest displacements at zero degrees. Is this true or is it due to sensor faults (or something else)?
R: In fact, these points are due to periods of missing data. There were removed in new version of these Figures.

5. Figure 21: the results of these analyses showed that the temperature measured at the deck sections presented different amplitudes and phase shifts when compared to the temperature measured at the arch sections. Indeed, the deck temperature is clearly lagging behind the arch temperature. However, by comparing it with the intraday variation of the natural frequencies (if available), which of the two most closely follow them? Related to this aspect: would the Authors suggest placing temperature sensors on the deck or at the arches, if they had to choose one of the two subsets?
R: As highlighted by the reviewer, and mentioned in the text over Figure 37, the temperature influences natural frequencies. Further studies need to be performed to select the adequate temperature signals to remove this environmental effect from modal identification. This needs to be done with a longer period of time of data. 
6. The paper proves that by using temperature measurements, it is possible to estimate the displacements in the expansion joints of the bridge accurately. Again, this was assessed by using sensors T1 to T4 (located on the deck). Does it mean that temperature sensors located on the deck are more predictive than temperature sensors located on the arches, for what concerns the displacements in the expansion joints?
R: The temperature of sensors T1 to T4 are as “smooth” as the displacements, which means they will be the most suitable for obtaining this correlation. Other sensors present more abrupt fluctuations that do not appear in the displacement signals, so it is concluded that the sensors T1 to T4 are more suitable for this correlation.
7. In the Introduction, the state-of-the-art review for bridge output-only monitoring can be extended. There are many recent studies, which also makes use of the same SSI-COV technique used here, both for masonry arch (https://doi.org/10.1002/stc.3028) or R.C. road bridges (https://doi.org/10.1016/j.measurement.2023.112451). These could be reported to provide some further context.
R: Taking into account the suggestions of all reviewers, the state-of-the art was improved, and one of the suggested papers was included in references list. 
Editorial and minor remarks:
1. Line 385, page 14; line 520, page 21: “Figure 385 23Error! Reference source not found.”
R: Correction done.
2. Since all authors belong to the same institution (FEUP), is not necessary to use different numbers in the Authors’ List.
R: Correction done.
3. Figure 2.b: by looking at it, it seems more like the accelerometer module of Figure 1.a than the sensor of Figure 2.a. Is it correct? From the text, it seems to understand that indeed the same external case was used. In that case, it may be useful to also portray the sensors of Figure 2.a as they appear inserted inside the polycarbonate box.
R: Indeed, the enclosure is the same. Temperature sensor (Figure 2b) is simpler and the electronic components are hidden. However, accelerometer module represented in Figure 1a was updated with legends to distinguish it from the others.

4. Figure 18 and elsewhere: please add the legend
R: The quality of many Figures was improved and updated (mainly graphs with stamps on x-axis). 
5. There are issues from Figure 16 to 22 (also Figure 33 and elsewhere). Specifically, it should be 2021 and 2022, not 0021 and 0022.
R: Correction done.
7. Overall, the content of the paper is very interesting and the data are clearly exposed but the quality of the Figure is a bit low. Some are grainy (e.g. Figures, 26, 37, 38, etc).
R: The quality of many Figures was improved as much as possible.
8. Page 18: it is not clear why ‘versus’ should be in italics.
R: Because this is not an English term, it is usual to put “versus” in italics.
9. Page 19 and elsewhere: all equations should be numbered.
R: Correction done.
10. Figures 5 to 10. This is a mere (and non-binding) stylistic suggestion but, if available, AutoCAD drawings could be better than the scanned original plans. 
R: No AutoCAD drawings are available. Figures are a scan of the original design (1950´s)…
Comments on the Quality of English Language
The English of the paper is overall good but it may use some grammar checking. There are a few mistakes and typos throughout the manuscript (e.g. page 18, line 456, ‘de’ instead of ‘the’).
R: Correction done. We assume that editor will make a final review to English.

Round 2

Reviewer 1 Report

Well revision and the paper is recommended to be published.

Reviewer 4 Report

 The authors have adequately faced the observations raised by this Reviewer.

Therefore, the manuscript can be accepted for publication, after careful proofreading.

the Quality of English is acceptable.